# Adenovirus Biology, Recombinant Adenovirus, and Adenovirus Usage in Gene Therapy

**DOI:** 10.3390/v13122502

**Published:** 2021-12-14

**Authors:** Maki Watanabe, Yuya Nishikawaji, Hirotaka Kawakami, Ken-ichiro Kosai

**Affiliations:** 1Department of Gene Therapy and Regenerative Medicine, Kagoshima University Graduate School of Medical and Dental Sciences, Kagoshima 890-8544, Japan; watanabe@m.kufm.kagoshima-u.ac.jp (M.W.); k6225679@kadai.jp (Y.N.); hirotaka_kawakami@yahoo.co.jp (H.K.); 2South Kyushu Center for Innovative Medical Research and Application, Kagoshima University Graduate School of Medical and Dental Sciences, Kagoshima 890-8544, Japan; 3Center for Innovative Therapy Research and Application, Kagoshima University Graduate School of Medical and Dental Sciences, Kagoshima 890-8544, Japan; 4Center for Clinical and Translational Research, Kagoshima University Hospital, Kagoshima 890-8544, Japan

**Keywords:** adenovirus, cancer immunotherapy, conditionally replicating adenovirus, cytokine, gene therapy, m-CRA, oncolytic virus, pluripotent stem cells, replication-defective adenoviral vector, survivin, vaccine

## Abstract

Gene therapy is currently in the public spotlight. Several gene therapy products, including oncolytic virus (OV), which predominantly replicates in and kills cancer cells, and COVID-19 vaccines have recently been commercialized. Recombinant adenoviruses, including replication-defective adenoviral vector and conditionally replicating adenovirus (CRA; oncolytic adenovirus), have been extensively studied and used in clinical trials for cancer and vaccines. Here, we review the biology of wild-type adenoviruses, the methodological principle for constructing recombinant adenoviruses, therapeutic applications of recombinant adenoviruses, and new technologies in pluripotent stem cell (PSC)-based regenerative medicine. Moreover, this article describes the technology platform for efficient construction of diverse “CRAs that can specifically target tumors with multiple factors” (m-CRAs). This technology allows for modification of four parts in the adenoviral E1 region and the subsequent insertion of a therapeutic gene and promoter to enhance cancer-specific viral replication (i.e., safety) as well as therapeutic effects. The screening study using the m-CRA technology successfully identified survivin-responsive m-CRA (Surv.m-CRA) as among the best m-CRAs, and clinical trials of Surv.m-CRA are underway for patients with cancer. This article also describes new recombinant adenovirus-based technologies for solving issues in PSC-based regenerative medicine.

## 1. Introduction

As of 26 October 2021, there were 22 cellular and gene therapy products (three products if cellular products are excluded) approved by the US Food and Drug Administration (FDA), including one oncolytic virus (OV) [1]. The last one and a half years (2020–2021) have proven to be a landmark in the history of pharmaceutical development due to worldwide demand for the rapid development of vaccines against COVID-19. Hence, the clinical development and approval of new modalities of gene therapy drugs, including mRNA and adenoviral vectors, have made remarkable progress worldwide. Consequently, the advantages of adenoviral vectors over conventional technologies, including efficacy, utility, and safety, as well as speed of development and commercial applications, have garnered widespread attention.

In this article, we review the biology of wild-type adenovirus, the construction of recombinant adenoviruses, including replication-defective adenoviral vectors and conditionally replicating adenoviruses (CRAs a.k.a. oncolytic adenoviruses). We also address the application of recombinant adenoviruses in cancer therapy and regenerative medicine.

## 2. Biology of Wild-Type Adenoviruses and the Principle behind Recombinant Adenoviruses

### 2.1. Biology of Adenoviruses

Adenoviruses are non-enveloped, icosahedral, 90–100 nm viruses with a linear, double-stranded DNA genome spanning 26–45 kb [2,3]. Adenoviruses are divided into seven species (A–G) and human adenoviruses are classified into more than 100 subtypes, including serotypes 1–52 and genotypes 53–103 [4,5]. In particular, serotypes 2 and 5 of species C, which often cause mild inflammation in cases of childhood infection of the upper respiratory tract, have been extensively studied and used in gene therapy technologies [6,7,8].

The cell surface receptors responsible for infections by individual adenoviruses vary among the different species. The coxsackie/adenovirus receptor (CAR) is the primary receptor for adenovirus species C, including serotypes 2 and 5 [9,10]. Attachment of adenovirus species C to the host cell surface is mediated by a high-affinity interaction between the knob domain of the adenovirus fiber and the CAR [11]. Subsequently, the interaction between cell surface integrins and an Arg–Gly–Asp motif, located in the viral capsid penton base, initiates virus internalization by receptor-mediated endocytosis [11,12]. Following entry, viral particles undergo a complex disassembly process to yield viral nucleocapsids that are transported along microtubules to the nuclear pore complex, where viral genomes are imported into the nucleus [13,14,15]. In contrast to retroviral infection, adenoviral infection does not depend on cell-cycle status and rarely involves the integration of viral genes into the host genome [16,17]. These features, which enable a broad range of infective cell types and reduced risk of chromosome mutagenesis, can provide great advantages in terms of utility and safety in in vivo gene therapy [18,19].

Serotype 5 adenoviral genome includes four early genes (E1–E4) and five late genes (L1–L5) that are transcribed before and after viral DNA replication, respectively (Figure 1) [6,20]. The early genes encode proteins that activate transcription of other viral genes and mediate viral DNA replication, while the late genes mostly encode viral structural proteins.

The immediate early gene (E1A) is first expressed after wild-type adenovirus infection and is the most important transcriptional activator for subsequent viral gene expression. Conserved region (CR) 2 of the E1A gene displaces retinoblastoma (Rb) proteins from E2F family transcription factors in infected cells, thereby inducing quiescent cells to progress into S-phase, which is essential for viral replication [21].

Two major proteins encoded by the E1B gene are the 55 kDa E1B55K and 19 kDa E1B19K [22]. E1B55K binds to the tumor suppressor p53 and promotes its degradation to prevent apoptosis induction, while E1B19K functions as an antiapoptotic factor. Together, the two proteins prevent the death of the infected host cell, thereby allowing viral replication to occur [6]. The late genes encode structural/capsid (e.g., penton, hexon, and fiber) and core (e.g., protein VII and protease) viral proteins [2,3,23]. Adenovirus also devotes many proteins, including E1B55K, E4orf3, E4orf4, E4orf6, and core protein VII, and utilizes a variety of mechanisms to inhibit the host’s DNA damage response (DDR); the DDR would otherwise restrict adenoviral replication [24].

### 2.2. Principle of Recombinant Adenoviruses

The E1A gene is the first gene expressed upon viral infection and is crucial for all subsequent viral gene expression. Hence, E1A deletion is commonly used to generate replication-defective adenoviral vectors [25]. In contrast, CRA have been generated by either mutating the E1A gene or by replacing the native E1A promoter with a cancer-specific promoter to alter E1A expression [26]. Since the adenoviral virion can package up to 105% of the wild-type genome length, some portions of viral genes that are not essential for virion formation, e.g., the E3 region, are often deleted to insert a therapeutic gene into a recombinant adenoviral genome [27,28]. The technology for constructing a replication-defective adenoviral vector was established in the 1990s. Currently, several commercial kits enable its experimental use without special expertise [29,30,31,32,33]. Therefore, this review does not describe detailed protocols for constructing replication-defective adenoviral vectors. Instead, it focuses on the construction and applications of CRA, which is currently an attractive tool for innovative cancer therapy. The characteristic features of a replication-defective adenoviral vector, CRA and CRA that can specifically target tumors with multiple factors (m-CRA) were described in Table 1.

### 2.3. CRA Construction

Research on OVs, which are recombinant viruses, including CRAs, that replicate in and kill cancer cells specifically, began in the late 1990s. As mentioned above, either mutation or altered expression of the E1 gene in adenoviral DNA is necessary to confer the property of cancer-specific viral replication on an adenovirus.

There are two strategies for generating CRAs [53,73]. The first involves deleting specific portions of the E1 genes [54,55,56,74]. The binding between the CR2 region of the adenoviral E1A protein and cellular Rb protein results in the transcription factor E2F being released from a preexisting cellular E2F-Rb complex. Consequently, E2F activates cell cycle-regulatory genes and induces quiescent cells to progress into S-phase, thereby providing a suitable environment for viral DNA synthesis [55]. This process is essential for adenovirus to replicate in quiescent normal cells, but is dispensable for replication in proliferating cancer cells, which have sufficient free E2F [75]. Accordingly, CRA that express mutant E1A will replicate in cancer cells that express free E2F but lack Rb; however, this replication is largely attenuated in normal cells with functional Rb. In contrast, early studies have suggested that the E1B55K protein, which binds to and inactivates p53 protein, protects infected cells from E1A-induced apoptosis and therefore permits efficient viral replication [24,76].

According to these ideas, a deletion in CR2 of the E1A gene and in the E1B55K gene within E1B would lead to selective adenoviral replication in cancer cells deficient in Rb and p53, respectively. Such mutant adenoviruses would not, however, replicate efficiently in quiescent normal cells with normal Rb and p53 proteins [57,58,77,78]. However, clinical trials involving E1B55K-deleted CRA have shown limited therapeutic effects for local and metastatic cancers. More importantly, recent studies have suggested that replication of E1B55K-deleted CRA is biologically more complex and not merely p53 status–dependent [76].

The second CRA generation strategy involves the transcriptional control of viral replication [59,60]. Since the E1A gene is essential for efficient adenoviral replication, replacing the native E1A promoter with a cancer-specific promoter leads to selective E1A expression, resulting in much higher levels of viral replication in cancer cells than in normal cells [61,79,80,81,82,83]. This strategy appears to depend on a simpler biological mechanism than the E1 mutation strategy. Hence, this transcription control strategy is promising if an ideal promoter, i.e., one that is both highly active and strictly cancer cell-specific, could be identified and used to regulate E1A and E1B transcription.

Given the above strategies, conventional CRAs have crucial technical limitations. First, no CRA has so far been developed that replicates efficiently in cancer cells yet is completely attenuated in normal cells. Conventional CRAs, particularly E1-deletion types, suffer from insufficient cancer cell specificity. Some CRAs may replicate in normal cells and cause cytopathic effects to some degree, even though their toxicity towards normal cells is attenuated compared with cancer cells [62,63,64]. Some studies have shown that the cancer specificity of CRAs is enhanced by two or three cancer-specific factors e.g., any combination of mutant E1A, mutant E1B, and upstream cancer-specific promoters, if not all of the above [65,84,85,86]. However, extensive and comparative studies on CRAs that are regulated with multiple cancer-specific factors were hampered by the lack of standardized methods to efficiently construct CRAs. The production of diverse types of highly modified CRAs in large numbers remains time-consuming and labor-intensive [84,85,86,87,88,89]. This fundamental issue must be solved for other type of OVs as well to maximize their potential.

### 2.4. CRAs That Can Specifically Target Tumors with Multiple Factors (m-CRA)

In order to solve this problem, we first developed a method for efficient construction of “CRAs that can specifically target tumors with multiple factors (m-CRA)”; the replication of these viruses can be simultaneously regulated by up to four independent factors of the E1 region: mutant E1A, mutant E1B, their transcriptional control by cancer-specific promoters, or all of the above [67]. In addition, our m-CRA technology makes the insertion of a therapeutic gene with a suitable upstream promoter into the m-CRA backbone feasible and efficient. The resulting m-CRA may be armed with either a cytotoxic or immune gene to enhance its anticancer effects, thereby adding distinct cytotoxic effects to the oncolytic effects of adenoviral proteins [64,67]. Moreover, the characteristic features of adenovirus, including the fiber region, which is a major determinant of infectivity, can be modified by replacement with other adenovirus backbones.

The experimental protocol was previously described in detail [67]. In brief, gene elements involving viral replication, therapeutic genes, and adenoviral backbones were introduced separately into three plasmids named P1, P2, and P3, respectively, which allows unrestricted construction and efficient fusion of individual elements (Figure 2). More than seven cancer-specific factors can be easily introduced into the m-CRA.

## 3. Application of Recombinant Adenoviruses

### 3.1. Replication-Defective Adenovirus-Based Cancer Gene Therapy and Vaccines

In vivo gene therapy for cancer using replication-defective adenoviral vectors has been studied extensively using both basic research and clinical trials, resulting in the conclusion that it is safe but ineffective. A common caveat in in vivo cancer gene therapy using any replication-defective vector is that while it may work for in vitro experiments, such vectors cannot physically transduce and kill all cancer cells when administered as in vivo therapy. An intracellular p53 gene therapy using replication-defective adenoviral vector was not approved by the FDA because of insufficient therapeutic efficacy [34]. Hence, a therapeutic gene that can kill both non-transduced and transduced cancer cells should be used for replication-defective vector-based cancer gene therapy.

In order to kill both non-transduced and transduced cancer cells with a replication-defective vector, we developed a combination gene therapy using a suicide gene and various cytokine genes delivered via a replication-defective adenoviral vector in the 1990s [35,36,37]. A suicide gene converts a nontoxic pro-drug into a toxic drug e.g., herpes simplex virus thymidine kinase converts non-toxic ganciclovir (GCV) to GCV-triphosphate, which causes actively dividing cells to die with a strong bystander effect, but not quiescent normal cells [38,39,40,41]. This effect results not only in tumor volume reduction, but also in the local release of tumor antigen that induces a cellular immune response. Locally, long-lasting secretion of cytokine proteins from transduced cells activates immune cells safely and efficiently [42,43,44]. Finally, the induced systemic antitumor cellular immunity abolishes non-transduced cancer cells both locally and in distant regions in animal experiments [35,36]. Likewise, a large number of immune genes, such as cytokine genes, are included in replication-defective adenoviral vectors for treating cancer [36,42,43,44,45,46].

This sort of strategy may provide some therapeutic benefits against both local and distant metastatic cancer cells in patients with certain types of cancer; however, the trend in cancer gene therapy has shifted almost completely from replication-defective vector-based immune gene therapy to OV immunotherapy (OVs armed with therapeutic genes). Nevertheless, early extensive studies on immune gene therapy have paved the way for current OV immunotherapies (see below).

In contrast, attempts at in vivo gene therapy using replication-defective adenoviral vectors for treating genetic disorders have failed due to the strong immunogenicity of adenovirus vectors. The representative case is a grievous incident, i.e., the death of the patient with ornithine transcarbamylase deficiency after an in vivo gene therapy clinical trial using replication-defective adenoviral vectors [47]. This is because adenoviral vectors exhibit multiple complex vector-host interactions and induce innate and adaptive immune responses [48]. To modulate adenoviral vector-induced immune responses, extensive genomic and chemical capsid modifications have been studied [48]. Helper-dependent adenoviral vector (HD-Ad), of which all viral genes except for cis-acting elements needed for genome replication and packaging are deleted, exhibited prolonged transgene expression and reduced toxicity [49]. Nevertheless, the acute inflammatory responses could be also induced after HD-Ad injection because the viral capsid of HD-Ad is identical to the first generation E1-deleted adenoviral vector. In this regard, methods to prevent viral vectors from being by recognized by innate and adaptive immune cells and to shield virions from the undesirable interactions with preexisting humoral immunity are potentially ideal [48]. Coupling capsid proteins to polymers and shielding attempts engineered protein coats showed promising results in vitro, but were ineffective in vivo [50,51]. Recently, a stealth layer based on hexon-binding single-chain antibody variable fragment was successfully used to shield vector particles from neutralizing antibodies [50]. In addition, the shield particles equipped with adaptor proteins achieved the predominant gene transduction to tumor cells in vitro and in vivo [50].

In contrast, such strong immunogenicity is useful not only for cancer gene therapy but also as a vaccination tool against infectious diseases. In fact, both adenoviral vector-based and mRNA-based gene therapy vaccines have just revealed not only more rapid development but also improved efficacy compared with conventional inactivated virus (protein)-based vaccines against COVID-19 [52].

### 3.2. History and Current Status of CRA-Based Oncolytic Virotherapy and Immunotherapy

As briefly described in Section 2.3, both extensive basic research and clinical trials involving OVs have been performed since the mid-1990s and early 2000s, respectively. However, early clinical trials on “OV, including CRA, without immune transgene” therapy, did not demonstrate significant efficacy for patients with intractable cancers, particularly those with metastatic cancers [66,90,91]. In order to enhance their anticancer effects by adding anticancer immunity to oncolysis, the development of OVs armed with therapeutic genes (OV immunotherapy) is being aggressively pursued worldwide [92]. The therapeutic mechanism has much in common with the underlying principle of our combination gene therapy using replication-defective adenoviral vectors (Section 3.1). It is as if the cancer-specific cytotoxicity of suicide gene therapy has been replaced by that of OV.

Amgen’s Talimogene laherparepvec (IMLYGIC^®^), which is herpes simplex virus type 1 carrying the granulocyte-macrophage colony-stimulating factor (GM-CSF) gene, was approved by the US FDA and the European Medicines Agency as a First-In-Class drug in 2015 [93,94]. Nevertheless, no OV immunotherapies, including IMLYGIC^®^, have yet to be declared a Best-In-Class product; a Best-In-Class drug would be able to eradicate all disseminated metastatic cancers [93]. It should, however, be noted that OV immunotherapy improved the therapeutic effects of immune checkpoint inhibitors by promoting intratumor T-cell infiltration in clinical trials [95]. Thus, the combination of OV immunotherapy and immune checkpoint inhibitor may become a standard therapy for patients with metastatic cancer.

### 3.3. Survivin Responsive m-CRA

Using our m-CRA platform technology, we screened the cancer specificity and transcriptional activity of several cancer-specific promoters, constructed diverse m-CRAs using the candidate promoters identified by the screen, and determined their virological characteristics [67]. In doing so, we identified survivin-responsive m-CRAs (Surv.m-CRA), in which wild-type E1A and mutant E1B (E1B55K-deletion) genes are regulated by the best region of the survivin promoter and the cytomegalovirus enhancer/promoter, respectively, as one of the best m-CRAs [68,69] (Figure 3). Survivin, a member of the inhibitor of apoptosis protein family, is expressed at high levels in cancer cells, but either very low or undetectable in normal cells [69,96]. Another group employed a telomerase reverse transcriptase (Tert)-responsive CRA in a human cancer trial although a partial response was observed in only one out of 15 patients [66].

Prior to this report, we identified the best region of the Tert promoter, which showed both the most strictly cancer-specific and the strongest activities, and which has been used to generate the best Tert-responsive m-CRA (Tert.m-CRA) [69,97]. Nevertheless, comparisons of viral properties and therapeutic potentials between Surv.m-CRA and Tert.m-CRA revealed that Surv.m-CRA is safer and more effective [69,97]. Another advantage of Surv.m-CRA is its high effectiveness towards cancer stem cells (CSCs), which are resistant to conventional chemo-radiotherapy [70,98,99]. Interestingly, past clinical studies have indicated a positive correlation between high survivin expression levels and poor prognosis, accelerated rate of recurrence, and increased resistance to therapy in patients with cancer [100]. Accordingly, Surv.m-CRAs killed all populations efficiently while also demonstrating increased therapeutic efficacy against CSCs [70].

Moreover, Surv.m-CRA-OCp, in which mutant E1B (E1B55K-deletion) expression is regulated by the osteosarcoma/prostate cancer-specific but weakly active osteocalcin promoter, significantly increased cancer specificity without decreasing anticancer effects against osteosarcoma and prostate cancer [68]. This finding suggests that cancer-specific or tissue-specific promoters, many of which have some cancer specificity but are relatively weak at driving transcription, are widely useful for also regulating mutant E1B expression, at least on the condition that E1A expression is regulated by a strictly cancer-specific and strongly active promoter. In contrast, replacing wild-type E1A with mutant E1A (24 amino acid deletion in the C2 region) downstream of the survivin promoter does not further increase the cancer specificity of Surv.m-CRA [69]. This finding suggests that producing the ideal m-CRA will depend more on altering the regulation of E1A expression than on mutating E1A. Likewise, platform technologies, including our m-CRA, that allow rapid generation of diverse next-generation OVs armed with transgenes downstream of suitable promoters, will be important [71].

We have conducted ICH (International Council for Harmonisation of Technical Requirements for Pharmaceuticals for Human Use) compliant GMP (Good Manufacturing Practice) drug production and GLP (Good Laboratory Practice) non-clinical studies and completed the First-In-Human investigator-initiated Phase I clinical trial of Surv.m-CRA-1 (Surv.m-CRA without a therapeutic gene) for patients with malignant tumors in bones and soft tissue. Both safety and the efficacy were good (results to be reported in the future), indicating that the results of basic research and preclinical studies were reproduced in this clinical application. Two clinical trials involving Surv.m-CRA-1 for patients with pancreatic cancer (Phase I/II) and malignant bone sarcoma (Phase II) are underway.

### 3.4. Application of Recombinant Adenoviruses to Pluripotent Stem Cell-Based Regenerative Medicine

Regenerative medicine based on human pluripotent stem cells (PSCs) is a potential, innovative therapy for intractable diseases. The application of recombinant adenoviruses has been studied by very few researchers, including the authors of this review, in comparison with recombinant retroviruses and lentiviruses, which integrate transgenes into the host genome [72,101,102,103,104]. In somatic cells, integration enables long-term expression of the inserted transgenes, but in undifferentiated PSCs, exogenous genes are silenced at a high frequency. The most critical issue is that integration poses significant safety risks as retrovirus-mediated ex vivo stem cell gene therapy causes a high incidence of leukemia in treated patients [105]. In contrast, adenoviral DNA and its transgenes very rarely integrate into the host chromosome, but the transgene persists as an episome in the nucleus, which confers a significant advantage in terms of clinical safety. In this respect, accurate knowledge of the properties of recombinant adenoviruses may facilitate the development of new biotechnologies to overcome the obstacle to human PSC-based regenerative medicine.

We first used a replication-defective adenoviral vector against PSCs, and developed an “adenoviral conditional targeting in stem cells” (ACT-SC) method that specifically separates any target cells from other types of differentiated cells and the remnant undifferentiated PSCs during and after in vitro differentiation [104]. In addition, we recently reported an optimized method for efficient adenovirus-based gene delivery to and expression in human PSCs [103]. Comparative studies on the activities of three representative ubiquitous promoters indicated that only the cytomegalovirus enhancer and beta-actin promoter allow robust transgene expression in human PSCs. Moreover, adenoviral vector infection of PSCs in single-cell suspension culture yields high gene transduction efficiency with low cytotoxicity, without causing loss of the undifferentiated state of the PSCs.

In contrast, despite their enormous potential, human PSC-derived cell transplantation therapies pose a risk of malignant transformation of undifferentiated remnants [102]. Conventional methods that establish safe human PSC master cell banks through gene expression analysis and genome-wide sequencing merely suppress and do not completely abolish tumorigenesis [102,106]. To eradicate tumorigenesis directly, we used gene therapy vector technologies, including recombinant adenoviruses for human PSC-based regenerative medicine. We developed three innovative strategies such as the ACT-SC method, a novel method for generating tumorigenic cell-targeting lentiviral vectors efficiently, and another novel method for applying m-CRA technology to kill tumorigenic PSCs specifically [30,72,101,102,103,104]. We found that the survivin promoter almost completely lost its activity in differentiated normal cells but was highly active in undifferentiated human PSCs as well as cancer cells [72,101]. Accordingly, in vitro infection experiments with human PSCs have indicated that Surv.m-CRA propagates vigorously and exerts a strong cytotoxic effect on undifferentiated and tumorigenic human PSCs; however, Surv.m-CRA did not damage normal cells that were differentiated from the same human PSCs. Moreover, in vivo experiments showed no tumor formation from the human PSCs that had been infected with Surv.m-CRA before being transplanted into animals. Because Surv.m-CRA safety was already verified in human clinical trials for patients with cancer as described in Section 3.3, obstacles to the usage of this technology on human patients may be largely mitigated.

## 4. Conclusions

More than thirty years have passed since the world’s first authorized gene therapy clinical trial in 1990, and data accumulated since from both basic and clinical studies are now blossoming into a host of practical applications. One important factor for success is the technology of recombinant viruses. In this respect, recombinant adenoviruses contribute to the development of innovative technologies, which will not only advance clinical cancer immunotherapy and vaccine development against infectious diseases, but also facilitate research on developmental biology and regenerative medicine using PSCs.

## Figures and Tables

**Figure 1 viruses-13-02502-f001:**
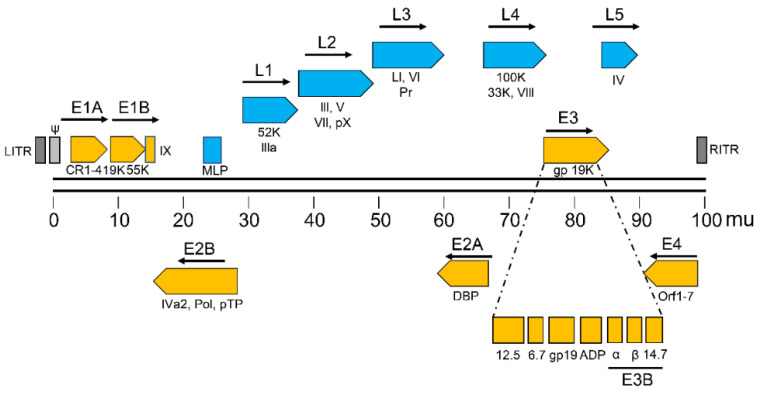
Schematic representation of the serotype 5 adenoviral genome. DNA strands are shown as a pair of lines. The number below the lines indicates the 0–100 map unit (mu) of the adenoviral genome.

**Figure 2 viruses-13-02502-f002:**
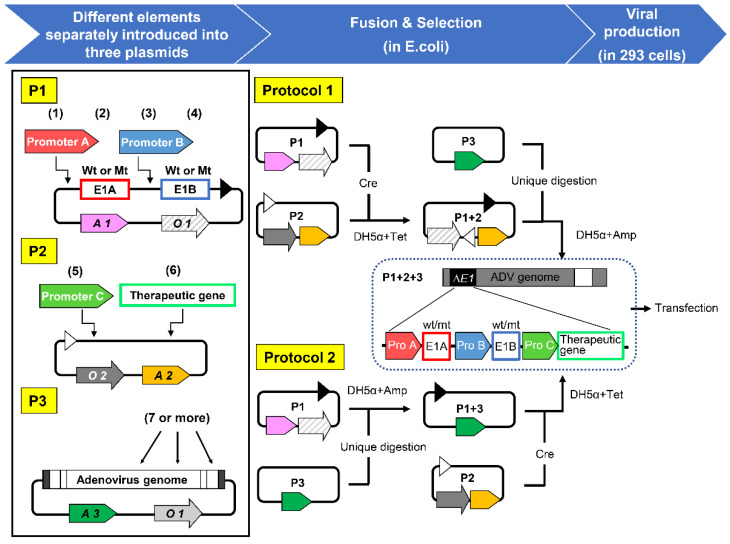
Constitution of m-CRA plasmids. Potentially, more than seven cancer-specific factors can be introduced into m-CRA. (1) Promoter A, which drives wild-type or mutant E1A expression. (2) Mutant E1A, which lacks an Rb-binding site (E1A∆24). (3) Promoter B, which drives wild-type or mutant E1B. (4) Mutant E1B, which lacks a p53-binding protein that is encoded by E1B55K (E1B∆55K). (5) Promoter C, which controls a therapeutic gene. (6) A therapeutic gene (seven or more can be incorporated). Modification of the adenoviral backbone is also possible, such as fiber modification to alter infectivity. A, antibiotic resistance gene; O, origin of replication.

**Figure 3 viruses-13-02502-f003:**
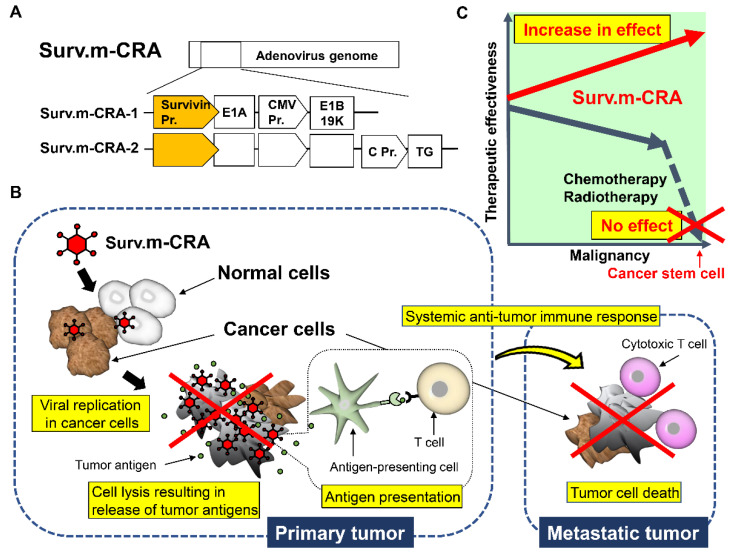
Therapeutic mechanism and characteristic features of survivin-responsive m-CRA (Surv.m-CRA). (**A**) The basic constitution of Surv.m-CRA. (**B**) Therapeutic mechanism of OV therapy, including Surv.m-CRA. (**C**) The putative therapeutic advantage of Surv.m-CRA against cancer stem cells in comparison to conventional chemo-radiotherapy.

**Table 1 viruses-13-02502-t001:** The characteristic features of recombinant adenoviruses.

Type of Recombinant Adenovirus	Modification of Adenovirus	Advantage	Disadvantage	Applicable Diseases and Therapies	References
Replication-defective adenoviral vector	E1A deletion	Robust but transient transgene expression and high immunogenicity applicable to cancer gene therapy and vaccines	No indication for genetic diseases because of short-term effect and safety concerns	Gene medicine for solid tumor Vaccination to prevent infectious diseases	[28,34,35,36,37,38,39,40,41,42,43,44,45,46,47,48,49,50,51,52]
Conditionally replicating adenovirus (CRA; oncolytic adenovirus)	A deletion in CR2 of the E1A gene and in the E1B55K gene within E1B Replacement of the native E1A promoter with a cancer-specific promoter	Predominantly replicating in and killing cancer cells Armed transgene enhancing therapeutic effects	Insufficient cancer-specific viral replication resulting in insufficient therapeutic effect of virotherapy alone	Virotherapy and immunotherapy for solid tumor	[21,25,53,54,55,56,57,58,59,60,61,62,63,64,65,66]
CRA that can specifically target tumors with multiple factors (m-CRA)	Simultaneously regulated by up to four independent factors of the E1 region Feasible to add a therapeutic gene with a suitable promoter and to modify adenoviral backbone in the step of viral constructions	Strictly cancer-specific and effective viral replication producing more potent anticancer effects and higher safety than the conventional CRAs More potently and safely enhancing therapeutic effects by a therapeutic gene with a suitable promoter and to alter infectivity		More potent and safer virotherapy and immunotherapy for solid tumor A new strategy to prevent stem cell-derived tumorigenesis in regenerative medicine	[67,68,69,70,71,72]

## Data Availability

Not applicable.

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
