# Peer review of "Adenovirus Biology, Recombinant Adenovirus, and Adenovirus Usage in Gene Therapy"

_viruses, 2021, doi:10.3390/v13122502_

Round 1
Reviewer 1 Report
The manuscript “Adenovirus Biology, Recombinant Adenovirus, and Adenovirus Usage in
Gene Therapy” by Watanabe et al. focuses on the utilization of adenoviruses in gene therapeutic applications. The authors give a brief introduction to adenovirus biology and introduce major concepts for constructing replication-deficient and conditionally-replicating Ad vectors.
The manuscript is presented with a clear structure and is comprehensibly written. It provides a comprehensive overview of state-of-the-art technologies and discusses current hurdles and limitations of these technologies. Thus, this review is of utmost importance for readers that are new to the field of Ad-mediated gene therapy and oncolysis.
I highly recommend the manuscript in its current state for publication. However, I encourage the authors to include a paragraph addressing current shielding strategies of the Ad capsids that will allow administration of Ad vectors via the bloodstream and might help to reduce Ad vector immunogenicity. Since the high immunogenicity of Ad vectors is explicitly mentioned (e.g. page 6, lines 210-219), this paragraph will strengthen the manuscript and may help the reader to obtain a better overview of the general field of Ad mediated gene therapy.
Reviewer 2 Report
The manuscript entitled "Adenovirus Biology, Recombinant Adenovirus, and Adenovirus Usage in Gene Therapy" written by Maki Watanabe et al is a review of the biology of wild-type adenoviruses, the methodological principle for constructing recombinant adenoviruses, therapeutic applications of recombinant adenoviruses, and new technologies in PSC-based regenerative medicine.
This manuscript provides a good overview of recombinant adenoviruses for gene therapy for cancer from the methodological aspects to the outcomes of the clinical applications. Therefore, I would recommend it for acceptance after the points listed below:
Recommendations for improvement:
- I would ask the authors to include a comparative table of different technologies to show their pros and cons as well as technical differences.
- The authors should provide the reference information at each text position where they mention their own studies, e.g., in the first paragraph of section 2.4. Personally, it would be encouraged for the authors to refer to their own works in the same way as others'.
- The authors should mention undesirable immune responses to adenovirus vectors and available circumvention strategies.
Minor comments:
- The number of the approved cellular and gene therapy products described in Introduction should be updated. As of November 13, 2021, 22 products are shown on the reference [1] page.
- Figure 1 appears to be an example of human adenoviruses (of species C?). Since there are significant differences in coding sequence composition and gene length between different genera and even between different species as summarized in 10.1099/vir.0.19697-0, the authors should provide which human adenovirus is mentioned in the figure and text.
- All "in vitro"s and "in vivo"s in the text should be italicized.
- Adding a "(see below)" or something appropriate to the end of the second paragraph of page 6 (line 209) would be helpful.
